# Combined Effects of Listening to Preferred Music and Video Feedback, during Warm-Up, on Physical Performance in Young Kickboxers

**DOI:** 10.3390/sports12050131

**Published:** 2024-05-14

**Authors:** Manar Boujabli, Nidhal Jebabli, Faten Sahli, Hajer Sahli, Makram Zghibi, Roland van den Tillaar

**Affiliations:** 1Research Unit: Sport Sciences, Health and Movement, Higher Institute of Sport and Physical Education of Kef, UR22JS01, University of Jendouba, Kef 7100, Tunisia; manarboujabli2019@gmail.com (M.B.); jnidhal@gmail.com (N.J.); sehli.feten@gmail.com (F.S.); sahlihajer2005@yahoo.fr (H.S.); makwiss@yahoo.fr (M.Z.); 2Higher Institute of Sport and Physical Education of Ksar Saïd, University of Manouba, Manouba 2037, Tunisia; 3Department of Sports Sciences and Physical Education, Nord University, 7600 Levanger, Norway

**Keywords:** fast tempo music, visual feedback, repeated roundhouse kick, RPE, feeling scale

## Abstract

Although studies have indicated that the prior use of video feedback and music listening separately improves physical performance and positive feelings in various sports, to our knowledge, no studies have investigated their combined effect in combat-sports-specific tasks. The aim of this study was to determine the combined effect of listening to preferred music and video feedback on aerobic and anaerobic performance in male kickboxers. In a counterbalanced crossover study design, twenty kickboxers underwent three kicking exercises under one of three conditions: (1) control condition, (2) combined listening to preferred music and video feedback, and (3) video feedback during 10-min of rope warm-up. Kickboxers performed a ten-second kicking test, multiple ten-second kick test, and progressive taekwondo test. The total number of kicks, fatigue index, heart rate, rate of perceived exertion, and feeling scale were measured. The combined music and video feedback condition improved the number of kicks with a better positive feeling scale (F ≥ 7.4, *p* ≤ 0.002, η_p_^2^ ≥ 0.28) than the video feedback and control conditions in all three kicking exercises, while the video feedback alone led to better kick performances and a better feeling scale than the control condition in the ten-second and multiple ten-second kicking tests (*p* ≤ 0.016). The combined listening to preferred music and video feedback condition was more effective at enhancing the positive feeling scale and repeated roundhouse kick performance. Future investigations should examine the application of video feedback and listening to music in various kickboxing tasks including punches and kicks.

## 1. Introduction

In order to optimize physical performance during competition, sports psychology is well established as a primary process in the preparation of athletes, during warm-up, through a variety of mechanisms including motivation, self-esteem, stress management, and concentration [1,2,3,4]. In this context, previous studies observed that warming up causes athletes’ sensorimotor networks to recalibrate, restoring their skills to a finely tuned state [5]. It also encourages players to improve their motor skills in accordance with the applicable biomechanical model and replicate these skills, for example, during the first meters of a race [6].

Therefore, self-regulation and motivational intervention during warm-up have been used by coaches to motivate an athlete or team before competition, such as the use of digital technology (video feedback) [7] and listening to music [8]. In fact, video feedback as a modern modality represents an essential factor alongside self-directed practice and the conscious control of attention [7].

Video feedback, using video clips, is considered one of the most important strategies before tests or competition to facilitate skill acquisition and improve physiological and psychological responses in order to achieve a better physical performance during competition [3,4,9].

Research on the use of video clips as a motivational strategy for an athlete during warm-up has shown an increase in pregame testosterone levels as well as small cortisol responses [3,9]. Also, watching video footage of successful skill execution before competition can enhance motivation and aggressive behavior (e.g., offensive moves in combat sports), and offers a better way to know the opponent’s flaws and physical performance [10].

This modality can be ensured in different ways and at different times [11]. In fact, previous studies have shown that using video feedback (e.g., expert modeling, self-modeling, or a combination of both) provides students and athletes the opportunity to identify their own mistakes during training [12,13] and physical education sessions [14,15]. Additionally, previous research has reported the effectiveness of self-assessment with video feedback on physical performance, including throwing tasks [16], sequential timing tasks [17], gymnastic tasks [18], as well as other specific tasks such as set shots in basketball [19] and trampoline jumping [20]. However, to our knowledge, no study has determined the effect of the self-assessment of a specific combat sports task via video feedback before exercise.

Besides video feedback, previous studies reported that listening to music during warm-up can also enhance physical performance [21,22,23,24]. In fact, the ergogenic effects of listening to music on physical performance, based on its motivational strategy, is attributed to the delayed perception of afferent signals related to neurological fatigue [25], improving muscular efficiency [26], neural activity [27], positive mood [22], attention [28], and self-efficacy [29].

For example, Ouergui et al. [24] observed that pre-selected warm-up music, using Brunel Music Rating Inventory 2 [30], enhances physical activity enjoyment and physical performances in taekwondo athletes. They reported that listening to music, using a fast tempo (140 beats/min) and high volume (80 dB) during warm-up, was more effective at improving physical performance and the frequency of kicking speed compared to conditions with less volume (+60 dB) or music with much higher tempo (200 beats/min) and control conditions. However, these responses may be affected by the quality of the athletes and the specificity of the task in which they were engaged [31].

Although both listening to music and video feedback have distinct benefits in sports, few studies have observed their combined effect on a physical test. To our knowledge, only Bayrakdaroğlu et al. [32] studied the combined effect of music and video feedback during warm-up on the physical performance of futsal players. They observed that listening to music and watching video feedback during the warm-up before a Wingate test promoted a significant improvement of physical performance compared to just listening to music and video feedback separately.

Nevertheless, it is unknown whether listening to music and video feedback during warm-up have an impact on the physical performance of repetitive fast discrete movements like kicking, which is much different from cyclical movements over a longer time such as the Wingate test. This type of feedback may have a potentially different impact on specific kickboxing techniques (kicking). Regarding kickboxing, this combat sport includes different styles, such as the light contact style, which involves two competitors directing punches (i.e., jabs, crosses, hooks, punches, flying punches, etc.) and kicks (i.e., front kicks, back kicks, roundhouse kicks, ax kicks, etc.) to the head and body [33,34]. The roundhouse kick and hook techniques are frequently used in kickboxing due to their high technical effectiveness during a fight [33]. Therefore, the aim of this study was to examine the effects of music and video feedback with different durations on kicking performance of amateur kickboxers. Based on the previous results of Bayrakdaroğlu et al. [32], we hypothesized that the combination of listening to music and video feedback, during warm-up, has more beneficial effects on physical performance and motivation than regular conditioning and video feedback in the field of combat sports as the combined condition stimulates both cognitive and motivational self-regulation.

## 2. Materials and Methods

### 2.1. Participants

The sample size of our study was determined using the power analysis program G*Power (version 3.1.9.3, University of Kiel, Kiel, Germany). The a priori power analysis was calculated using the F-test family (repeated one-way ANOVA measures) and a related study that examined the effects of combined music and video feedback on physical performance [32]. The power analysis was performed using our primary outcome, an assumed power of 0.90, an alpha level of 0.05, and a medium effect size of 0.33 for a minimal sample size of 17 participants [32]. Therefore, we recruited additional kickboxers (n = 20) to account for potential dropouts. Accordingly, a total of 20 amateur male kickboxers (light contact kickboxers; age: 17 ± 2 years; height: 1.69 ± 0.12 m; body weight: 62 ± 12 kg; BMI: 21.1 ± 1.6) were recruited. The inclusion criteria were to have kickboxers (light contact specialty) who had regularly competed at a national level for at least 3 years (4 ± 1 years) but had not been successful in winning any medals. The kickboxers’ regular training sessions were 3 times per week. Also, none of the kickboxers underwent any weight loss procedures or had any medical conditions or a history of specific musculoskeletal injuries before and during experimentation period.

All participants and their parents gave their written agreement after being informed of the study’s protocol. Before the commencement of the tests, the protocol was fully approved by the local Ethics Committee of the Higher Institute of Sports and Physical Education of Kef (UR22JS01; 03/2024), University of Jendouba, Tunisia. The current study was carried out in accordance with the Declaration of Helsinki in 2013.

### 2.2. Procedure

A week before the testing session, kickboxers had two familiarization sessions to become well acquainted with the experimental procedures. During the familiarization sessions, video recordings were made of each kickboxer and for each test in order to show them during the warm-up conditions of the experimental sessions. In addition, all kickboxers selected their preferred songs.

During the experimental sessions, kickboxers performed in a counterbalanced crossover study design under the following conditions: (1) warm-up with music and video feedback test session, (2) warm-up with video feedback session, and (3) warm-up with no music or feedback (control session), before performing a physical test (Figure 1). Each session started with a standardized 10 min warm-up rope jump, in an area of 2 × 2 m, to facilitate the viewing of feedback videos using a tablet. To avoid any change in the metabolic cost during the warm-up, kickboxers exercised by rope jumping at 50% of their age-predicted of maximum heart rate (%HRmax) using an individual heart rate monitor (Polar FT4; Polar Electro Oy, Kempele, Finland). Two minutes after the warm-up, kickboxers performed (1) frequency kicking for ten seconds, (2) five series of 10 s of frequency kicking, or (3) a progressive taekwondo test.

All the tests carried out are based on the repetition of the roundhouse kick technique, which is frequently used in kickboxing due to its high technical efficiency during competition [33]. The 10 s and multiple 10 s tests were performed on the same day with two hours in between, while the progressive taekwondo test was performed on a separate day. Each test day was 48 h apart. Thus, a total of nine test sessions were conducted with each kickboxer. To avoid any potential effect of diurnal variation on performance, all tests were performed at the same time of day (±1 h) in the same gym (temperature 18–23 °C). The same investigators and the same instructions were provided in each condition.

#### 2.2.1. Video Feedback

Video feedback of each kickboxer was recorded during the familiarization sessions. Indeed, the video camera was placed 3 m from the kickboxer’s side (sagittal view). The height of the kickboxers was used to calibrate the height of the camera, using an adjustable tripod, to track their movements. For this experiment, we used the GoPro4 session camera, which allowed us to record videos at a resolution of 1080p (1920 × 1080, 16:9) and 48 frames per second (Fps). The GoPro4 session camera (GoPro, Inc., San Mateo, CA, USA) was connected to a laptop (Dell Inspiron 15 3000 series; Dell Inc) using a USB cable. During the three tests, video clips for each individual trial were recorded on the laptop. Video clips were recorded without any editing and without sound.

In the experimental sessions, kickboxers received their video feedback during the warm-up period before each test. Kickboxers saw their own video clip with slow playback speed (0.5×, 0.5 times slower than normal speed). Videos automatically replayed for 10 min of the jump rope warm-up. The projection of the video clips was carried out by a video projector device (Epson EB-FH52) connected to a laptop via an HDMI cable and broadcast using the Kinovea video analysis software (version 0.9.5, Bordeaux, France). Epson EB-FH52 projects clear videos onto the wall in high resolution (full HD 1080p). No verbal feedback or verbal guidance was given to the kickboxers by the investigator. Also, conversation between the investigator and the kickboxer was kept to a minimum during the experimental sessions.

#### 2.2.2. Listening to Music

All kickboxers selected their preferred songs. After measuring the tempo of all songs with the “Edjing Mix” app (version 6.45.00, Android, MWM, Boulogne-Billancourt, France), we found that all kickboxers selected songs with a fast tempo (>140 beats/min). Moreover, the same music volume (moderate music, 70 dB) and the same mp3 player Bluetooth Wireless Earphone (Beats Power beats Wireless, Beats Electronics Ttc, US) were used during experimental sessions.

#### 2.2.3. Single and Multiple 10 s Kicking Tests

In the frequency kicking for 10 s and the multiple 10 s kicking test, the kickboxer stood in front of a bag and performed the maximum number of roundhouse kicks by alternating between the right and the left leg during 10 s of testing. In the multiple 10 s test, this was repeated for five sets with 10 s of rest between each set [35]. The total number during the 10 s test and the multiple 10 s test was used as a performance parameter. In the multiple 10 s test, besides the total number of kicks over all sets, the best set (the greatest number of kicks recorded during a single set) and fatigue index were also recorded. The fatigue index corresponded to the relative power decrement and was calculated according to the following formula:Fatigue index (%) = [1 − ((total number of kicks over all sets)/(set with most kicks × 5))] × 100.

During the multiple 10 s kicking test, peak and mean heart rate (HRpeak, HRmean) were recorded using an individual heart rate monitor (Polar TF4, Polar Electro, Kempele, Finland). Pilot data from 20 participants collected on two different days were used to determine the reproducibility of the test and showed ICCs ≥ 0.881 for all these parameters.

#### 2.2.4. Progressive Taekwondo Test

In the progressive taekwondo test, the kickboxer started in the same position and with the right leg, as in the other two tests, with six kicks for 100 s, alternating legs at the first stage, followed by gradually increasing by 4 kicks with each new stage, while the duration of the stages decreased proportionally (Table 1) [36].

Each athlete kept pace until exhaustion. The total number of kicks corresponding to the last completed stage was used for further analysis. During the tests, no verbal encouragement was given. In each stage, the investigator provided information about the number of kicks that the kickboxer had to do. Also, the time of each stage and the interval of kicks were controlled by a computerized beep that initiated the kick. During the test, peak and mean heart rates were recorded using a heart rate monitor (Polar team 2, Polar Electro Oy, Finland). Pilot data from 20 participants collected on two different days were used to determine the reproducibility of the test (ICC = 0.917).

#### 2.2.5. Perceived Exertion and Feeling Scale

After each test, ratings of perceived exertion (RPEs) of legs were measured on a 10-point scale (CR-10 Borg-scale) ranging from 0 (no exertion) to 10 (maximum exertion) [37]. In addition, feeling state, measured on an 11-point scale ranging from +5 (very good) to −5 (very bad) with a midpoint of 0 (neutral), was used to assess the affective state of the participants immediately after the tests [38]. The instructions were as follows: “try to inform us, by a number, on your inner feeling without concerning yourself with feelings of physical stress, effort, and fatigue”.

### 2.3. Statistical Analyses

Data were expressed as means and standard deviations (SD). Normality of data was assessed and confirmed using the Kolmogorov–Smirnov test. The reliability of all tests was assessed by the intra-class correlation coefficient (ICC). Kicking performance and physical and psychological responses were assessed using a one-way analysis of variance (ANOVA) with repeated measures for each variable. In addition, for physical performance of the multiple 10 s test, a 3 (condition) × 5 (sets) ANOVA with repeated measures was performed to assess the effect of feedback for every set of kicks. If a difference was significant, then a Holm–Bonferroni post-hoc test was computed. The effect size was evaluated by partial eta squared (η_p_^2^), where η_p_^2^ < 0.06 represents a small effect, 0.06 ≤ η_p_^2^ < 0.14 a medium effect, and η_p_^2^ ≥ 0.14 a large effect [39]. Statistical analyses were performed using SPSS software version 27.0 (SPSS, Inc., Chicago, IL, USA). The level of statistical significance was set at *p* ≤ 0.05.

## 3. Results

Kicking performance was significantly affected in all three tests by feedback (F ≥ 7.4, *p* ≤ 0.002, η_p_^2^ ≥ 0.28). The post-hoc comparison revealed that during the 10 s and multiple 10 s tests, the significantly highest number of kicks was performed under the combined music and video feedback condition, followed by the video feedback condition, and the significantly lowest number of kicks under the control condition (Figure 2). During the progressive taekwondo test, all kickboxers completed all levels (510 kicks) under the combined music and video feedback condition, while during both control and video feedback conditions, not all kickboxers achieved all levels (six athletes did not). Therefore, a significantly higher number of kicks was only achieved in the combined condition (*p* = 0.014) compared with the other two conditions.

When evaluating the multiple 10 s kicking test per set, a significant effect of feedback condition, set number, and interaction effect were found (F ≥ 4.1, *p* ≤ 0.001, η_p_^2^ ≥ 0.18). The post-hoc comparison showed that under the music and video feedback condition, in every set, more kicks were performed compared to the other two conditions (*p* ≤ 0.021), and the most kicks were performed during the first set in all conditions. In all conditions, the number of kicks decreased significantly in set two (*p* < 0.01). In the combined music and video feedback condition, the number of kicks continued to decrease significantly every set, while in the video condition it only significantly decreased between sets 3 and 4. In the control condition, the numbers of kicks did not decrease significantly after set 2 (Figure 3).

When comparing the psychological responses after the tests, in all three tests a significant effect of feedback on both RPE and feeling scale was found (F ≥ 4.1, *p* ≤ 0.024, η_p_^2^ ≥ 0.18). The post-hoc comparison indicated that the RPE was significantly lower in all three tests after the warm-up music in combination with video feedback was used compared with the other two conditions. Meanwhile, the feeling scale was significantly higher in all tests with music and video feedback, followed by video feedback alone, and the lowest scores in all tests were found in the control condition (Table 2).

No significant effect of feedback condition was found for the mean (F = 0.13, *p* = 0.88, η_p_^2^ < 0.01) or peak heart rate (F = 2.96, *p* = 0.064, η_p_^2^ = 0.135) during the progressive taekwondo test or for the fatigue index during the multiple 10 s kicking test (F = 1.8, *p* = 0.176, η_p_^2^ = 0.09, Table 3).

## 4. Discussion

The aim of this study was to investigate the effects of combined video feedback and listening to preferred music during warm-up, compared with just video feedback or no feedback, on kicking performance for different durations in amateur kickboxers. The main findings were that the combined music and video feedback condition had a better effect on kick numbers and psychological responses than the video feedback and control condition in all three kicking tests, while the video feedback alone resulted in better kick performances and better feeling scales than the control condition in the 10 s and multiple 10 s kicking tests.

As hypothesized, the combined music and video feedback during the warm-up had better results in terms of kicking performance and psychological responses than the video feedback alone and control condition, which is in accordance with Bayrakdaroğlu et al. [32], who investigated the effect of music and video feedback during warm-up on the physical performance of futsal players. They observed that the combination of listening to preferred music (>140 beats/min, 70 dB) and video feedback before the Wingate test, during the warm-up, significantly improved power output indices better than video feedback and control conditions.

The performance improvement was probably due to the fact that this condition stimulates both cognitive and motivational self-regulation. As previous studies have shown, personal video feedback allows subjects to self-correct technical movements, thereby reinforcing the learning of technical aspects that could subsequently improve physical performance [40,41], which was also evident from the higher number of kicks in the 10 s and the multiple 10 s kicking tests (Figure 2).

Also, using video clips while warming up can increase testosterone levels and trigger smaller cortisol responses in order to enhance motivation and physical performance during test and competition [3,4,9].

Just listening to fast (140 beats/min) and loud (80 dB) music during warm-up has been shown by Ouergui et al. [24] to enhance kicking performance during the 10 s and multiple 10 sec kicking test in taekwondo athletes. Other studies [22,24] also indicated that the use of preferred music with a fast tempo (>140 beats/min) presents an effective strategy to improve physical performance, probably due to the motivational self-regulation of the music [22,23]. Due to the earlier findings of the positive effects on performance by the use of music during the warm-up, this condition was not included in the present study. According to previous studies [25,27], preferred music can promote improved physical performance depending on systemic physiological changes during exercise, which are related to neural activation through increased brain activity, particularly the motor cortex region, with an increased autonomic response.

The increased kicking performance was accompanied by a lower RPE and higher feeling scale during the combined condition, while absolute intensity was similar, as shown by the heart rates (Table 2). This may also be due to the dissociation effect of music. Indeed, dissociation as measured by RPE can be used to distract from feelings of discomfort or fatigue during exercise at different intensities [40], and preferred fast tempo music (>140 beats·min^−1^) has been described to exacerbate attention diversion and dissociation [33]. In practice, reducing the RPE helps individuals to tolerate higher intensity levels during exercise [1], which was the case in the present study.

Another important outcome from this study was that the combined music and video feedback condition, as well as just video feedback, improved the feeling scale after the tests compared to the control condition. These results are consistent with previous research that showed a significant improvement of positive mood and motivation during aerobic and anaerobic exercise after listening to music or video feedback [24,42,43,44]. However, it seems that a combination of both music and video feedback enhanced the feeling scale even more after the tests.

To our knowledge, no study has evaluated the combined effect of listening to preferred music and video feedback on emotional responses during physical activity. However, separate studies have reported that music or video feedback results in a greater positive feeling during exercise [43,45], increasing intrinsic motivation [2] and mood state [24,43]. Therefore, it can be concluded that listening to preferred music improved physical performance more than the video feedback condition during warm-up.

The current study reveals important information regarding the optimization of performance through the use of combined music and video feedback during warm-up. However, this study has limitations that should be considered. Firstly, the present study focused only on the effect of listening to preferred music and video feedback before testing. Thus, the results cannot be generalized to other separate studies on video feedback and music during exercise. Secondly, the condition with preferred music was not carried out in this research, which could be more useful to evaluate the differentiation between video feedback and music. However, it is necessary to take into consideration the increase in the number of repetitions of the tests, under different conditions, in relation to the parasitic learning effect, especially when the specific tests have the same technical execution. Thirdly, the progressive taekwondo test is perhaps unrepresentative for the assessment of maximal aerobic performance because many athletes performed all levels under the condition of combined music and video feedback, with only six kickboxers who did not complete all the test levels during the other two conditions. Therefore, future studies will be necessary to modify the number of levels of this test.

As a practical application, to ensure better self-regulation of kickboxing techniques with motivational intervention, it is preferable for kickboxers to use their video feedback and preferred music during warm-ups. Also, through video feedback, experienced kickboxers can self-regulate their individual technical errors and immediately engage in critical analysis of their own execution of the movement without the coach describing them. However, listening to preferred music and receiving video information from personal videos feedback improved kickboxers’ motivation and reflected positively on their performance during exercise.

## 5. Conclusions

It was concluded that the effect of listening to preferred music combined with video feedback during warm-up was more effective at enhancing positive mood and physical performance of repeated roundhouse kicks in aerobic and anerobic tests than video feedback alone or a regular warm-up in young male kickboxers. Further investigations are needed to verify whether music and video feedback could affect performance and psychophysiological responses in other tests more specific to kickboxing, for both male and female genders.

## Figures and Tables

**Figure 1 sports-12-00131-f001:**
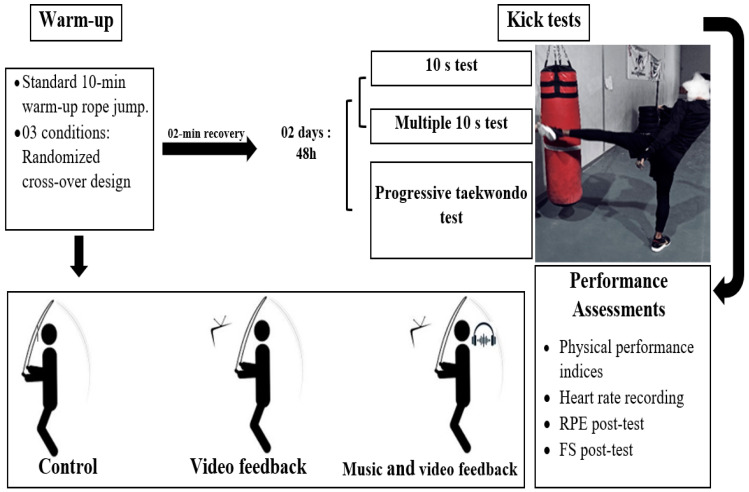
Experiment design. RPE, ratings of perceived exertion; FS, feeling state.

**Figure 2 sports-12-00131-f002:**
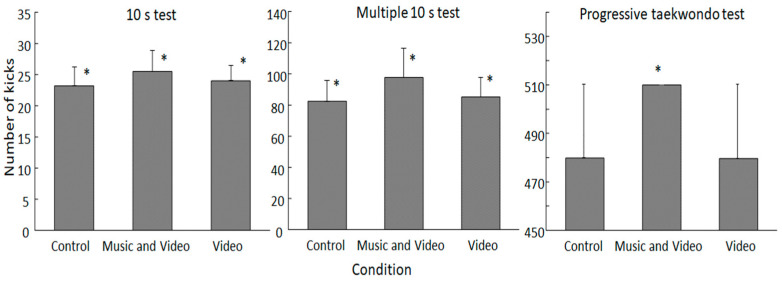
Mean (±SD) total number of kicks during the three kicking tests. * indicates a significant difference with all other conditions at the *p* < 0.05 level.

**Figure 3 sports-12-00131-f003:**
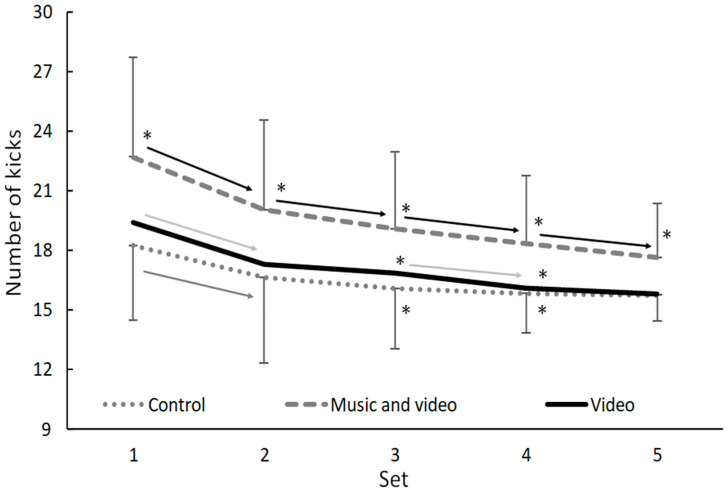
Mean (±SD) number of kicks during each set of the multiple 10 s kicking test. * Indicates a significant difference from all other conditions on a *p* < 0.05 level. The arrow indicates a significant decrease from one set to the next for a particular condition.

**Table 1 sports-12-00131-t001:** Duration and number of kicks per stage during the specific progressive kicking exercise.

Stage	Duration (s)	Number of Kicks	Frequency of Kicks (kick/min)
**1**	100	6	3.6
**2**	84	10	7.1
**3**	77.1	14	10.9
**4**	73.3	18	14.7
**5**	70.9	22	18.6
**6**	69.2	26	22.5
**7**	68	30	26.5
**8**	67.1	34	30.4
**9**	66.3	38	34.4
**10**	65.7	42	38.4
**11**	65.2	46	42.3
**12**	64.8	50	46.3
**13**	64.4	54	50.3
**14**	64.1	58	54.3
**15**	63.9	62	58.2

**Table 2 sports-12-00131-t002:** Mean (± SD) RPE and feeling scale scores after each test in each condition.

	Video Feedback	Music and Video Feedback	Control	*p*-Value (η_p_^2^)
**RPE**				
10 s test	2.6 ± 0.51	2.3 ± 0.47 *	2.6 ± 0.51	0.034 (0.31)
Multiple 10 s test	8 ± 0.89	7.5 ± 0.83 *	8.1 ± 0.91	0.002 (0.50)
Progressive taekwondo test	9.6 ± 0.60	8.7 ± 0.8 *	9.5 ± 0.60	0.002 (0.49)
**Feeling scale**				
10 s test	3.1 ± 0.87 *	3.6 ± 0.82 *	2.7 ± 0.81 *	0.005 (0.45)
Multiple 10 s test	4.2 ± 0.77 *	4.6 ± 0.60 *	3.6 ± 0.94 *	<0.001 (0.60)
Progressive taekwondo test	3.2 ± 0.77 *	3.8 ± 0.79 *	2.6 ± 0.94 *	<0.001 (0.58)

η_p_^2^: partial eta squared; * indicates a significant difference with all other conditions on a *p* < 0.05 level.

**Table 3 sports-12-00131-t003:** Mean (±SD) average and peak heart rate during the specific progressive test and the fatigue index of the multiple 10 s kicking test per condition.

	Video Feedback	Music and Video Feedback	Control
**Average heart rate (beats/min)**	178 ± 4	178 ± 5	178 ± 5
**Peak heart rate (beats/min)**	193 ± 3	192 ± 3.1	193 ± 3.2
**Fatigue index (%)**	10.9 ± 7.6	13.0 ± 6.3	9.9 ± 5.9

No significant differences between any of the conditions on a *p* < 0.05 level.

## Data Availability

The raw data supporting the conclusions of this article will be made available by the authors, without undue reservation.

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
