# Peer review of "Combined Effects of Listening to Preferred Music and Video Feedback, during Warm-Up, on Physical Performance in Young Kickboxers"

_sports, 2024, doi:10.3390/sports12050131_

Round 1

Reviewer 1 Report

Comments and Suggestions for Authors

This manuscript aims to determine the combined effect of listening to preferred music and video feedback on aerobic and anaerobic performance in male kickboxers. The subject is very interesting and the manuscript is well structured. I have the following suggestions.

1. "kick-boxers" or "kickboxers", please keep it consistent.

2. The Introduction lacks content regarding "kickboxers".

3. Since the 10s test and multiple 10s tests were conducted on the same day, there is controversy over whether the test results were caused by the intervention. The correct procedure is to conduct the 10s test and multiple 10s tests separately after each of the three interventions, meaning that the intervention should be re-applied before each test.

4. The aim of this study was to determine the combined effect of listening to preferred music and video feedback on aerobic and anaerobic performance in male kickboxers. However, there were no aerobic performance indicators, as the progressive Taekwondo test is actually a muscle endurance test.

Author Response

Dear Reviewer 1,

Thank you for giving us the opportunity to submit a revised draft of the manuscript “Combined effects of listening to preferred music and video feedback, during warm-up, on physical performance in young kickboxers” for publication in Sports. We appreciate the time and the effort that you dedicated to providing feedback on our manuscript and we feel grateful for the insightful comments and valuable suggestions. We have incorporated most of the suggestions made by you and the reviewers and carefully considered your concerns. All changes are highlighted in yellow throughout the revised manuscript. Please see below for a point-by-point response to the reviewers’ comments and concerns.

This manuscript aims to determine the combined effect of listening to preferred music and video feedback on aerobic and anaerobic performance in male kickboxers. The subject is very interesting and the manuscript is well structured. I have the following suggestions.

  1. "kick-boxers" or "kickboxers", please keep it consistent.

Response : Corrected. Thank you

  1. The Introduction lacks content regarding "kickboxers".

Response : Thanks for your suggestion. Content concerning kickboxing has been added in the introduction part (line 92-97):” Regarding kickboxing, this combat sport includes different styles, such as the light contact style, which involves two competitors directing punches (i.e. jabs, crosses and hooks, jabs, punches flying punches, back kicks) and kicks (front kicks, roundhouse kicks, ax kicks, etc.) to the head and body [33, 34]. The roundhouse kick and hook techniques are frequently used in kickboxing due to their high technical effectiveness during the fight [33].”

  1. Ouergui, I., Hssin, N., Franchini, E., Gmada, N., & Bouhlel, E. Technical and tactical analysis of high level kickboxing matches. Int. J. Perf. Anal. Spor. 2013, 13(2), 294–309. https://doi.org/10.1080/24748668.2013.11868649
  2. Ouergui, I.; Davis, P.; Houcine, N.; Marzouki, H.; Zaouali, M.; Franchini, E.; Bouhlel, E. Hormonal, physiological, and physical performance during simulated kickboxing combat: Differences between winners and losers. Int. J. Sports Physiol. Perform. 2016, 11(4), 425–431. https://doi.org/10.1123/ijspp.2015-0052

  1. Since the 10s test and multiple 10s tests were conducted on the same day, there is controversy over whether the test results were caused by the intervention. The correct procedure is to conduct the 10s test and multiple 10s tests separately after each of the three interventions, meaning that the intervention should be re-applied before each test.

Response : Thank you for your comment. Firstly, the 10s test and the multiple 10s tests have the same technical conditions of performance (the difference appears in the number of repetitions) which do not need to be carried out separately and in two distinct sessions. Second, a short period of intense exercise (10 seconds) requires only 2 to 5 minutes of sufficient recovery to significantly reduce muscle fatigue (Froyd et al., 2013, 2020).

  • Froyd, C., Millet, G. Y., & Noakes, T. D. (2013). The development of peripheral fatigue and short‐term recovery during self‐paced high‐intensity exercise. The Journal of physiology, 591(5), 1339-1346.
  • Froyd, C., Beltrami, F. G., Millet, G. Y., MacIntosh, B. R., & Noakes, T. D. (2020). Greater short-time recovery of peripheral fatigue after short-compared with long-duration time trial. Frontiers in physiology, 11, 525289.

  1. The aim of this study was to determine the combined effect of listening to preferred music and video feedback on aerobic and anaerobic performance in male kickboxers. However, there were no aerobic performance indicators, as the progressive Taekwondo test is actually a muscle endurance test.

Response : Thank you for your comment. According to the validation article by Sant'Ana et al. (2019), the taekwondo progressive test presents itself as a valid tool for assessing the power and aerobic capacity of Taekwondo athletes based on direct comparisons with a treadmill test. They reported that Vo2peak values from the taekwondo progressive test were similar to those obtained from direct measurements of Vo2peak on a nonspecific test performed on a treadmill. Therefore, the aerobic indicator was calculated from the total number of kicks as well as the heart rate responses (Average heart rate, Maximum heart rate) during the progressive taekwondo test.

  • Sant'Ana, J., Franchini, E., Murias, J. M., & Diefenthaeler, F. (2019). Validity of a taekwondo-specific test to measure VO2peak and the heart rate deflection point. The Journal of Strength & Conditioning Research, 33(9), 2523-2529.

Reviewer 2 Report

Comments and Suggestions for Authors

This is an outstanding applied study that contributes to the field. The following minor considerations are offered in hopes of enhancing the ms. Further:

-The Abstract should start with a brief summary as to how the current study contributes to the relevant literature (fills a gap). Moreover, in the Abstract, when comparing the experimental conditions on outcome measures, try to keep the description of measures the same throughout (i.e., across conditions so comparisons can be briefly assessed). For example, “more psychological responses,” “better feeling scale,” and “enhanced mood” – results of these outcomes should be reported for all conditions in a similar way. It can be assumed that the feeling scale was used across the three conditions, so how did participants in each of these conditions do relative to the other on this measure, specifically. Is a “feeling scale” the same as “enhanced mood” or “psychological response”

-On line 42, music needs a supportive study reference, and the Discussion would benefit from a brief examination of an earlier study specific to the performance effects of music:   

Miller, A., & Donohue, B. (2003). The development and controlled evaluation of athletic mental preparation strategies in high school distance runners. Journal of Applied Sport Psychology, 15, 321-334.

Comments on the Quality of English Language

See my comments.

Author Response

Dear Reviewer 2,

Thank you for giving us the opportunity to submit a revised draft of the manuscript “Combined effects of listening to preferred music and video feedback, during warm-up, on physical performance in young kickboxers” for publication in Sports. We appreciate the time and the effort that you dedicated to providing feedback on our manuscript and we feel grateful for the insightful comments and valuable suggestions. We have incorporated most of the suggestions made by you and the reviewers and carefully considered your concerns. All changes are highlighted in green throughout the revised manuscript. Please see below for a point-by-point response to the reviewers’ comments and concerns.

Comments and Suggestions for Authors

This is an outstanding applied study that contributes to the field. The following minor considerations are offered in hopes of enhancing the ms. Further:

-The Abstract should start with a brief summary as to how the current study contributes to the relevant literature (fills a gap). Moreover, in the Abstract, when comparing the experimental conditions on outcome measures, try to keep the description of measures the same throughout (i.e., across conditions so comparisons can be briefly assessed). For example, “more psychological responses,” “better feeling scale,” and “enhanced mood” – results of these outcomes should be reported for all conditions in a similar way. It can be assumed that the feeling scale was used across the three conditions, so how did participants in each of these conditions do relative to the other on this measure, specifically. Is a “feeling scale” the same as “enhanced mood” or “psychological response”

Response: The abstract part has been corrected according to your recommendations. Thank you.

-On line 42, music needs a supportive study reference, and the Discussion would benefit from a brief examination of an earlier study specific to the performance effects of music:   

Miller, A., & Donohue, B. (2003). The development and controlled evaluation of athletic mental preparation strategies in high school distance runners. Journal of Applied Sport Psychology, 15, 321-334.

Response: Supporting study reference was added. Thank you

Round 2

Reviewer 1 Report

Comments and Suggestions for Authors

Dear authors, thank you for your reply.

Reviewer 2 Report

Comments and Suggestions for Authors

Looks great.